# Public Health Residents’ Anonymous Survey in Italy (PHRASI): Study Protocol for a Cross-Sectional Study for a Multidimensional Assessment of Mental Health and Its Determinants

**DOI:** 10.3390/ijerph20032003

**Published:** 2023-01-21

**Authors:** Alessandro Catalini, Clara Mazza, Claudia Cosma, Giuseppa Minutolo, Valentina De Nicolò, Veronica Gallinoro, Marta Caminiti, Angela Ancona, Lorenzo Stacchini, Nausicaa Berselli, Eleonora Ferrari, Fabrizio Cedrone, Vincenza Gianfredi

**Affiliations:** 1Department of Biomedical Sciences and Public Health, Università Politecnica delle Marche, 60100 Ancona, Italy; 2Department of Public Health, Experimental and Forensic Medicine, University of Pavia, 27100 Pavia, Italy; 3Department of Health Sciences, University of Florence, 50134 Florence, Italy; 4Department of Health Promotion, Mother and Child Care, Internal Medicine and Medical Specialties, University of Palermo, 90127 Palermo, Italy; 5Department of Public Health and Infectious Disease, Sapienza University of Rome, 00185 Rome, Italy; 6Department of Medicine and Surgery—Sector of Public Health, University of Perugia, 06100 Perugia, Italy; 7School of Hygiene and Preventive Medicine, Vita-Salute San Raffaele University, 20132 Milan, Italy; 8Department of Biomedical, Metabolic and Neural Sciences, University of Modena and Reggio Emilia, Via Campi, 287, 41125 Modena, Italy; 9Hospital Management, Local Health Authority of Pescara, 65100 Pescara, Italy; 10Department of Biomedical Sciences for Health, University of Milan, Via Pascal, 36, 20133 Milan, Italy; 11CAPHRI Care and Public Health Research Institute, Maastricht University, 6211 Maastricht, The Netherlands

**Keywords:** mental health, surveys and questionnaires, schools, public health, healthcare personnel, protocol study, cross-sectional design

## Abstract

The COVID-19 pandemic has evolved into a severe psychosocial crisis affecting patients, their relatives, friends, and healthcare professionals. In Italy, public health residents (PHRs) remain essential to the national response to the pandemic. To assess their mental sphere, the “Public Mental Health” working group of the medical residents’ Assembly of the Italian Society of Hygiene and Preventive Medicine has designed the Public Health Residents’ Anonymous Survey in Italy (PHRASI). This is a nation-wide cross-sectional study based on an 88-item self-administered voluntary survey that evaluates how sociodemographic variables are associated with mental issues, including wellness, eating disorders, sleeplessness, alcohol misuse, depression, and anxiety. Data will be gathered by disseminating a Google Forms link across the Assembly network of medical residents. All PHRs enrolled in a four-year program in one of the Italian postgraduate schools of public health will be qualified as participants. PHRASI aims to draw a comprehensive and detailed picture of the mental health state of Italian PHRs. PHRs are a significant group of healthcare professionals that may serve as a future benchmark for developing and enacting regulations intended to support the mental health of healthcare professionals.

## 1. Introduction

The World Health Organization (WHO) defines mental health as a “state of well-being in which the individual realizes his or her own abilities, can cope with the normal stresses of life, can work productively and fruitfully, and is able to make a contribution to his or her community” [1].

Throughout our lives, multiple determinants may combine to protect or undermine our mental health. The social determinants of mental health encompass five key domains [2], including demographic (sex, age, and ethnicity), economic, and socio-cultural domains that act and interact with each other at both the individual and community levels. It is therefore likely that a combination of two or more of these determinants connotes populations that are highly vulnerable and at increased risk of adverse mental health outcomes [3]. Mental disease is not a single pathology but, on the contrary, is a composite condition, including several mental disorders and psychosocial disabilities, as well as other mental states associated with significant distress, impaired functioning, or risk of self-harm. Generally speaking, mental disorders are characterized by a clinically significant disturbance in an individual’s cognition, emotional regulation, or behaviour. Moreover, mental disorders are increasing worldwide. According to the Global Burden of Diseases, Injuries, and Risk Factors Study (GBD) 2019, mental disorders have increased by 48.1% between 1990 and 2019 [4]. Among these disorders, depression and anxiety represent the two most common mental disorders and two of the leading causes of burden worldwide (ranked 13th and 24th in disability-adjusted life-years, respectively), with relatively higher prevalence estimates and disability burdens than many other diseases. The persistently high prevalence of these disorders, in addition to bipolar disorder and eating disorders, is of particular concern because they not only compromise health per se but also increase the risk of other health outcomes, such as suicide (ranked as the 18th leading cause of mortality in GBD 2019) [5].

As mentioned above, mental illnesses are thought to result from complex interactions among genetic, biological, psychological, and environmental factors (including social and cultural ones) [6]. Evidence has shown that mental disorders often arise in an individual with a genetic makeup that presents a predisposition. In addition, many experts believe that the impaired regulation of neurotransmitters may contribute to mental disorders. This vulnerability, combined with other stressful events (family difficulties and violence) [7], or unhealthy life habits (physical inactivity, increased alcohol consumption and obesity) can lead to mental disorders [8]. Among the stressful events, the COVID-19 pandemic has had a strong impact on people’s mental health [9]. According to the GBD 2020, this has led to a 27.6% increase in cases of major depressive disorder (MDD) and a 25.6% increase in cases of anxiety disorders worldwide in 2020 [9].

Moreover, the COVID-19 pandemic seems to have affected the mental sphere in many dimensions: the onset of sleep problems [10], increase in harmful use of alcohol [11], worsening of the risk of new onset, recurrence, and relapse of eating disorders [12], and the outbreak of burnout and work-related stress especially among health care workers [13]. Indeed, during the COVID-19 pandemic, one of the work-categories most affected by mental disorders were frontline workers, especially in the health sector [14]. They have played a crucial role in fighting the pandemic and saving lives, while being subjected to extreme workloads, risks of becoming infected and spreading the infection to families and communities and witnessing the death of patients. Despite several studies exploring the impact of the COVID-19 pandemic on the mental health of the general public, or on certain subgroups of the population (such as younger or more vulnerable people, including healthcare workers in general) [15,16,17,18,19,20,21], none of these have been conducted among medical residents, and more specifically among public health residents. Moreover, previous studies have mainly explored a single mental health condition or a limited number of potential associated factors.

The Public Health Residents’ Anonymous Survey in Italy (PHRASI) study was designed in order to investigate how several social determinants and aspects of one’s lifestyle are associated with different mental health outcomes. The PHRASI study aims at comprehensively investigating the mental sphere of the aforementioned health workers: from anxiety to insomnia, from their relationship with food to the sense of coherence. In the current paper we will present and discuss the methodology applied in designing and developing the PHRASI study.

## 2. Materials and Methods

PHRASI is a nation-wide cross-sectional study designed to investigate various dimensions of mental health and its determinants among Italian Public Health Medical Residents (PHRs).

### 2.1. Design of the Survey

In December 2021 the “Public Mental Health” working group of the medical residents’ Assembly of the Italian Society of Hygiene and Preventive Medicine started to design the study. A questionnaire was developed based on a discussion among the working group participants, expert consultation and literature review. The working group was formed by Italian medical residents in Public Health. The participants of the group are the authors of the manuscript and those listed in the acknowledgements. The experts involved were recruited among the universities to which the members of the working group belonged. Concerning the literature review, the working group brainstormed which mental health dimensions and its determinants should be investigated with the questionnaire. A bibliographic search on PubMed was carried out to find related tools for those dimensions of mental health for which there was a unanimous willingness, on the part of the group, to investigate. The keywords used included the name of the dimension to be investigated, and the words “questionnaire” and “validation”. For each tool the following information was collected in an ad hoc spreadsheet developed in Excel (Microsoft Excel^®^ for Microsoft 365 MSO, USA, 2019): name of the questionnaire, validation article reference, if this validation was pertinent to the Italian population, whether its use was free of charge, the number of items, estimated time of compilation, aim, further references and notes. The tools collected were then discussed in the working group to select the most suitable to be included in the survey. Tool selection was made on the basis of the following criteria: validity and reliability inferred from the validation articles, shortness of completion, possibility of self-administration and frequent use in scientific literature. A questionnaire on work-related stress not yet validated was also included.

Further meetings of the working group led to the selection of socio-demographic variables to be used as predictors.

The selected tools and socio-demographic questions were then composed into a single questionnaire on Google Forms (©2022 Google, Mountain View, CA, USA).

From 30 May to 13 June 2022, the questionnaire was accessible only to the members of the working group, who carried out test compilations in order to check for errors in the structure or content of the questionnaire.

### 2.2. Eligibility

Eligible for participation will be all of the approximately 1600 medical residents enrolled in the four-year-course of any of the Italian postgraduate public health schools.

### 2.3. Data Collection

The Google Forms link will be shared through the network of the medical residents’ Assembly of the Italian Society of Hygiene and Preventive Medicine. The official mailing list and chats of the Assembly will be used to spread the link among PHRs on a large scale. Subsequently, representatives of each Italian postgraduate public health school will be contacted individually to incentivize them to fill out the questionnaire and promote it among their colleagues. Participation will be anonymous and voluntary with no incentives to increase participation. Participants will have the option to stop completing the questionnaire and resume it later. Since it is not possible to know how many PHRs will receive the link, the response rate cannot, at this time, be calculated. The questionnaire will be available to complete for about one month.

### 2.4. Questionnaire

The questionnaire will be accompanied by a cover letter (see Appendix A) informing the recipient about the aim of the study, the management of the collected data and the voluntariness and anonymity of the questionnaire. The obligation to answer all questions has been enabled on Google Forms. The final version of the survey consists of 88 items, with an estimated duration of completion of about 12 min. The survey is based on two main sections: (i) sociodemographic characteristics, and (ii) mental health and its determinants.

The first section consists of three sub-areas: (i) personal data, (ii) traineeship-related data, and (iii) work-related data. The personal data section gathers questions about age, gender, region of residence, whether the respondent is engaged in a stable relationship, cohabitation, and offspring. The traineeship-related section comprehends questions about region of traineeship, course year in the postgraduate school, whether they are off-site, whether they are a commuter, their willingness to work in the current work/training place after completion of the postgraduate course, their simultaneous attendance of two traineeships, and their intention to repeat the test to enter a different postgraduate school/general practitioner course. The work-related section asks PHRs for information about whether they have a contract of employment compatible with the postgraduate school, if they do then how many hours per week, if the hours spent within the work contract were considered also valid for the completion of postgraduate school, and their capacity to make ends meet with their own income.

The second section, intended to investigate different dimensions of mental health and its determinants, consists of the following ten different tools, as specified below.

Wellbeing will be assessed through the Italian versions of Self Related Health-5 (SRH-5) [22,23] and the WHO-5 wellbeing index [24,25]. SRH-5 is a validated single-item tool with five possible responses structured on a Likert scale that measures the current perceived general health. The WHO-5 index is a five-item validated questionnaire aimed at assessing the current mental wellbeing. Each item assigns 0 to 5 points. The final score is obtained by summing the points for each question and multiplying by four. It ranges from 0 to 100 where 0 represents the worst imaginable wellbeing and 100 represents the best imaginable wellbeing.

Physical activity will be assessed using the Italian and short version of the International Physical Activity Questionnaire (IPAQ) [26,27]. IPAQ is a validated questionnaire, comprising seven questions and tested for use in adults (age range of 15–69 years) asking the type (vigorous, moderate, walking) and the quantity (days per week and time per day) of the physical activity in the past seven days. In order to describe the overall level of activity, the questions are designed to provide separate scores for walking, moderate-intensity activity, and vigorous-intensity activity, as well as a combined total score. The last question is about the time spent sitting on a weekday and is not included in the calculation of the final combined score. However, two extra questions were added: the first to assess the intensity of walking activity (intense, moderate or slow) and the second to distinguish between time spent sitting during working days and weekend days. Neither question influences the score, and consequently they do not impact on the validity of the questionnaire.

Alcohol abuse will be assessed through the administration of the Italian translation of Alcohol Use Disorders Identification Test -c (AUDITc) [28]. AUDITc is a validated three-item short version of the AUDIT questionnaire. Each item attributes 0 to 4 points. A final score equal to or greater than five for males and equal to or greater than four for females indicates a possible risky consumption of alcohol.

The Italian translation of the Work-Sense of Coherence Questionnaire (work-SoC) will be used to assess a work-related sense of coherence, which is defined as the perceived comprehensibility, manageability and meaningfulness of an individual’s current work situation [29]. Work-Soc is a validated nine-item tool that asks to express for nine corresponding aspects and indicate on a seven-grade Likert scale how the person finds his work and job situation in general. The result is made up of three end points which represent respectively the state of work-related perceived comprehensibility, manageability and meaningfulness. The Cronbach’s alpha calculated in the validation study was 0.83.

Work-related stress will be assessed with a new and non-validated questionnaire: the Work-Related Stress Questionnaire (WRSQ) [30]. This is a 13-item tool whose validation will be the object of a subsequent paper.

Eating disorders will be investigated with the Italian version of Sick, Control, One, Fat, Food Test (SCOFF Test) [31,32]. The SCOFF test is a five-item validated screening tool, often used in the primary care setting, that asks for an answer of “yes” or “no” to each question. The presence of at least two “yes” is indicative of a possible eating disorder and deserves further study. The Cronbach’s alpha in the Italian validation study was 0.64.

Insomnia will be assessed with the Italian version of the Insomnia Severity Index (ISI) [33,34]. This is a brief validated instrument designed to evaluate the severity of night-time and daytime components of insomnia. It consists of seven questions. The seven answers are added to obtain a total score. From 0 to 7 insomnia is not clinically significant, 8–14 means subthreshold insomnia, 15–21 moderate severity clinical insomnia and 22–28 severe clinical insomnia. The Cronbach’s alpha of this questionnaire in its validation study was 0.76.

Anxiety will be evaluated with the Italian translation of the Generalized Anxiety Disorder-7 (GAD-7) [35]. GAD-7 serves as a screening tool and an assessment of the intensity of the four most prevalent anxiety disorders’ symptoms (generalized anxiety disorder, panic disorder, social phobia and post-traumatic stress disorder). The validated questionnaire asks how often in the past two weeks the participant has suffered from seven different anxiety-related problems. An overall score ≥ 3 indicates the possible presence of an anxiety-related disorder.

Finally, depression will be assessed with the Italian version of the Patient Health Questionnaire 9 (PHQ-9) [36,37], the major depressive disorder module of the full PHQ. This is a validated tool used for the screening of depression and in the evaluation of the severity of symptoms in mental health and general medical settings. Each of the nine items, corresponding to symptoms of major depressive disorder, assigns 0 (not at all) to 3 (nearly every day) points, providing a 0–27 severity score. The final question, which asks “How difficult have these problems made it for you to do your work, take care of things at home, or get along with other people?” is not scored but provides a good indication of the patient’s overall impairment. The Cronbach’s alpha in the validation study was 0.89.

### 2.5. Sample Size

The sample size was calculated in order to obtain statistically significant results for depression, as prevalence of depressive symptoms is the primary outcome of the study. A precision/absolute error (*d*) of 5% and type 1 error (*Z_1−α/2_*) of 5% (*p* < 0.05) were set. Given the lack of information on depression prevalence (*p*) among PHRs in Italy, data from the meta-analysis by Mata DA et al. were used. According to this, 28.8% of medical residents had depressive symptoms [38]. Therefore, applying the formula provided by Charan and Biswas for cross-sectional studies [39]:$$\mathrm{sample}\;\mathrm{size}=\frac{Z_{1-\alpha /2}^{2}p(1-p)}{d^{2}}$$
a sample size of 315 was calculated.

The same calculations were made considering anxiety as the secondary outcome. Since there are no studies on the prevalence of anxiety in Italian resident physicians, research in the literature has highlighted a prevalence (*p*) of this mental health condition of about 15% in resident physicians from other countries [40,41]. Applying the same formula, the sample size required is 196.

### 2.6. Ethical Considerations, Patient Information and Written Informed Consent

This study does not require the approval of an ethical committee because the questionnaire data will be totally anonymous, making it impossible to identify and harm any respondent. Moreover, neither drugs nor medical devices will be prescribed/administered. As a result, the responses will be collectively examined while taking into account Italian and European regulations governing the management of personal data [42,43,44,45].

In more detail, anonymity will be granted by not asking the name, surname and date of birth. In order to retrieve information about the geographical area of participants while still preserving anonymity, the region in which the participants work will be asked instead of the university to which they belong. To further protect PHRs’ identities, it will be possible to choose to not specify a gender.

The cover letter of the questionnaire will inform the participants that the data will be used only for scientific purposes, will be archived for a maximum of five years and will be accessible only by the members of the “Public Mental Health” working group of the medical residents’ Assembly of the Italian Society of Hygiene and Preventive Medicine. Google Form will be used only to spread the questionnaire and to collect data. All the collected data will then be downloaded and stored in a professional computer and protected by a password.

It will be possible to complete the questionnaire only after the participants declare that they understand the methods and purpose of the study by clicking on “I give the consent” option in reference to the processing of personal data. Participants who disagree will be redirected to a thank you message.

## 3. Results

The discussion among the working group participants, the expert consultation and literature review led to the identification of 34 tools investigating various dimensions of mental health and its determinants. Among these, 10 were selected and included in the final questionnaire. Their characteristics are summarized in Table 1.

In brief, nine out of ten tools were previously validated. Although the Italian translations of the questionnaires were found for all of them, validation studies for these Italian versions were found for only six tools. All ten of the tools were free of charge. The number of items of each tool varied from one (Self-related Health) to 13 (Work-related Stress Questionnaire). The estimated time for completion varied from 5 s for the Self-related Health to 3 min and 30 s for the International Physical Activity Questionnaire.

## 4. Discussion

Nowadays, the Western world is facing an increase in mental diseases [48,49], impacted heavily by the COVID-19 pandemic [50], and mostly affecting healthcare personnel, who are more susceptible to anxiety, depression, burnout and insomnia [51,52]. In more detail, previous evidence has shown that during a pandemic, medical doctors are more prone to develop both negative personal (hypervigilance, fatigue, difficulty sleeping, unhappiness and dejection) and professional symptoms (feeling lonely, doubting their professional employment and work-related distress) [53]. The impact of COVID-19 on medical doctors’ mental health has drawn attention on the importance of psychosocial wellbeing in patients’ management [54], especially because physicians’ psychosocial wellbeing is often neglected [55,56,57]. Moreover, the novelty of the virus, with general unpreparedness in diagnosis and treatment, as well as the rapid spread of the infection, with thousands daily cases and hundreds of deaths in each country, not only caused mental health distress of healthcare workers, but also led to healthcare staffing shortages. In this context, the Italian government allowed for the official engagement of public health residents in health services, in order to identify and manage infected cases and ensure effective preventive strategies. However, despite the high burden of the disease, as well as the complexity and multidimensional aspects of aetiology [58,59], there is a dearth of information and paucity of data regarding residents’ mental health and specific associated factors.

Considering this, the current study aims at comprehensively assessing several social determinants and lifestyle factors potentially associated with and influencing PHRs’ psychological wellbeing in a broad view. Numerous psychological components will be explored, such as, for instance, anxiety, depression, eating disorders, and sense of coherence; additionally, several sociodemographic, work-related and financial characteristics will be studied.

On one hand, we hypothesize no significant association between mental wellbeing and some demographic characteristics such as age and educational level, because these are commonly shared among the target population. On the other hand, we expect statistically significant differences in the level of physical activity [60,61], geographical residency or variables associated with the work environment as has been suggested in previous studies where different target populations have been assessed [62,63]. Moreover, we hypothesize statistical differences in characteristics specifically connoting PHRs, such as the simultaneous attendance of two traineeships and the intention to repeat a test to enter a different postgraduate school/general practitioner course. In this context, and considering that residents spend most of their time at work, we hypothesize that the two aforementioned characteristics could have a significant impact on residents’ mental health, burdening them with instability in their present and uncertainty about the future.

Research data for the development of evidence-based approaches are essential to reduce the negative consequences of the pandemic on psychological health [64]. The results of our study should demonstrate the primary mental-health determinants of PHRs in order to favour the adoption of prevention measures. 

The achievement of the awareness of PHRs’ mental health status might entail several future interventions. Firstly, if needed, the development of psychological and psycho-social support for PHRs, even using telehealth tools to make these services more accessible; moreover, the introduction of psychotherapeutic support centres in universities that lack them; lastly, evidence-based interventions to promote mental health and wellbeing among PHRs and workers [65,66].

To the best of our knowledge, no previous studies have been conducted on residents’ mental health in Italy or in Europe. In order to fill this gap, PHRASI was designed to investigate mental health status in residents, focusing on PHRs, who represent those directly involved in health promotion and prevention. This research could be considered unique and innovative in its genre and particularly relevant to the assessment of Italian PHRs’ mental wellbeing. In addition to its originality, this study presents various strengths.

The first strong point of the study is the mandatory nature of the survey’s answers, allowing us to gain complete and trustworthy data and to eliminate the possibility of missing information. Moreover, most of the questions of the survey are closed-ended and participants are provided with options to choose a response from them. Compared with open-ended questions, this is a more convenient and standardized way to obtain information, because every respondent selects from the same answer pool. This allows us to reduce missing data and errors of compilation [67].

Exploring a wide range of domains (wellbeing, depression, anxiety, alcohol abuse, physical activity, work-related stress, work-sense of coherence, eating disorders, and insomnia) through validated and widely used questionnaires, this study offers a complete view of the different dimensions of mental health and gives a global and comprehensive evaluation of the mental wellbeing of the studied population.

Moreover, this study allows us to gain a representative picture of the PHRs characteristics, ensures the nation-wide generalizability of data to both Italian PHRs’ mental wellbeing and to the broader PHR population, which faces the same sources of stress.

This study can be defined as cost-effective because it uses an online and free platform to build and share the questionnaire. Furthermore, no questionnaires have been printed on paper, reducing the environmental impact of the research project.

### 4.1. Practical Implications

This study should provide a multidimensional assessment of mental health in Italian PHRs and its determinants. We expect to collect relevant information on different dimensions of mental health (including depressive symptoms and anxiety) and its determinants (both in terms of sociodemographic data, lifestyle factors and work-related aspects) among PHRs.

The awareness of the sociodemographic factors associated with the presence of mental health symptoms might entail several future interventions on public health residencies in order to minimize the occurrence of PHRs’ dissatisfaction. Information gathered may be used by policy makers in the framing of concrete strategies to create a satisfying work–life balance in order to improve residents’ gratification and mental well-being.

### 4.2. Limitations

Our major limitation includes the cross-sectional design of the study, in which outcomes and exposures regarding mental health issues are measured at the same time. Effectively, we will take a “snapshot” of PHRs’ mental wellbeing at one specific point in time, regardless of when they first developed a mental disease [68]. Therefore, this study design allows us to investigate the prevalence of mental health diseases in PHRs but not to assess their incidence, neither temporality nor causality. However, the results of our study might be relevant considering the gap in knowledge.

The study is based on self-reported measures and not on diagnoses based on clinical data. Therefore, there may be a discrepancy between the real values and those reported by the participants. Moreover, some tests used in the questionnaire (for instance the one on depressive symptoms) are not diagnostic tools. They are rather classified as screening tests to help clinicians make a diagnosis. Having clinically depressive symptoms does not equate to having a major depressive disorder; however, previous studies have found a high level of sensitivity and specificity of the PHQ-9 [69,70,71,72]. Therefore, we are confident that the misclassification rate will be low. Moreover, it should be noted that, although alcohol consumption is explored by means of the AUDIT questionnaire, it does not ask about the consumption of other drugs, which could be highly relevant. 

Direct self-reports might provide underestimates of the true prevalence, as the results might be potentially biased by socially desirable responses [73], where respondents may have a tendency to answer in accordance with social norms rather than truthfully [74]. Being competent in the matter, PHRs might try to make a favourable impression by falsely reporting things about themselves or answering according to what would be expected of a doctor of public health and preventive medicine [75]. 

The adoption of a completely anonymous questionnaire, with online administration and no interviewer–interviewed contact reduces this bias [76]. However, despite the precautions put in place, socially desirable bias is subtle and difficult to mitigate and therefore a certain bias of response should be expected.

The nonresponse bias might also be present since the cross-sectional survey comes with an emailed questionnaire.

## 5. Conclusions

PHRASI is the first study with the aim of assessing the status of Italian PHRs’ mental wellbeing and, despite its limitations, has the potential to successfully bridge the gap related to residents’ mental health during and after the COVID-19 pandemic. 

It is important to note, however, that future researchers should explore other domains of mental health in order to improve the quality and effectiveness of this research.

The results of this study will be relevant because they will provide information regarding PHRs’ mental health and, as a consequence, might represent a starting point for the monitoring of mental health over time. Moreover, PHRASI results might represent a benchmark against which other (national or international) studies can be compared. In addition, a cross-sectional study is a track opener for further, more expensive, studies, such as case-control and cohort studies. Lastly, whilst this survey has been specifically tailored for use in public health residents, it could be adapted to be used in different resident populations.

## Figures and Tables

**Table 1 ijerph-20-02003-t001:** Characteristics of the selected tools to include in the final questionnaire.

Name of the Questionnaire	Validation Article Reference	Validation for Italian Population	Free of Charge	Number of Items	Estimated Time of Completion (min)	Aim	Further References and Notes
General Anxiety Disorder -7	Spitzer RL et al., 2006 [35]	not present	Yes	7	0:45	Screening for generalized anxiety disorder	Italian version available at https://www.ecfs.eu/sites/default/files/general-content-files/working-groups/Mental%20Health/GAD7_Italian%20for%20Italy.pdf (accessed on 4 November 2022)
Patient health Questionnaire -9	Kroenke K et al., 2001 [36]	Mazzotti E et al., 2003 [37]	Yes	9	1:00	Screening of depression	Italian version available at https://www.demenzemedicinagenerale.net/images/test/PHQ-9_Ok_20-2-2016.pdf (accessed on 4 November 2022)
Alcohol Use Disorders Identification Test -c	Bush K et al., 1998 [46]	not present	Yes	3	0:30	Screening for dangerous use of alcohol	Translated italian version by the National Institute of health at https://www.epicentro.iss.it/alcol/apd2014/scheda%20test%20audit%20c.pdf (accessed on 4 November 2022)
International Physical Activity Questionnaire	Lee PH et al., 2011 [26]	Minetto MA et al., 2018 [27]	Yes	7	3:30	Measuring type and quantity of physical activity	http://www.societaitalianadiendocrinologia.it/public/pdf/questionario_ipaq.pdf (accessed on 4 November 2022)
World Health Organisation-5 Wellbeing index	Topp CW et al., 2015 [47]	Nicolucci A et al., 2004 [25]	Yes	5	0:30	Measuring perceived wellbeing	High negative predictive value for the diagnosis of depression; Italian version available at https://www.psykiatri-regionh.dk/who-5/Documents/WHO5_Italian.pdf (accessed on 4 November 2022)
Sick, Control, One, Fat, Food Test	Morgan JF et al., 2000 [31]	Pannocchia L et al., 2011 [32]	Yes	5	0:30	Screening for eating disorders	Italian version available at https://www.sisdca.it/public/pdf/Scoff-Q.pdf (accessed on 4 November 2022)
Work-Sense of Coherence Questionnaire	Vogt K et al., 2013 [29]	not present	Yes	9	1:00	Investigating sense of coherence dimensions (Comprehensibility, Manageability and Meaningfulness) in the work-related context	Related to health literacy. Ongoing validation study in an Italian population.
Insomnia Severity index	Bastien CH et al., 2001 [33]	Castronovo V et al., 2016 [34]	Yes	7	0:50	Evaluating the perceived severity of insomnia symptoms	There are numerous questionnaires for sleep quality, but this one is short, so it could be more applicable to our goal
Work-related Stress Questionnaire	not validated	not present	Yes	13	1:30	Screening for work-related stress	https://pubmed.ncbi.nlm.nih.gov/32614365/ (accessed on 4 November 2022)
Self-related Health	Ware JE Jr, Sherbourne CD 1992 [22]	Cislaghi B, Cislaghi C 2019 [23]	Yes	1	0:05	Brief evaluation of perceived general health	/

## Data Availability

Authors can be contacted for information about data concerning the tools selected and those rejected to be part of the final questionnaire.

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
