# Peer review of "Public Health Residents’ Anonymous Survey in Italy (PHRASI): Study Protocol for a Cross-Sectional Study for a Multidimensional Assessment of Mental Health and Its Determinants"

_ijerph, 2023, doi:10.3390/ijerph20032003_

Round 1

Reviewer 1 Report

Congratulations for your work, I think it is a very interesting project, but I would still like to clarify and change some important points, which are attached in a separate file.

Author Response

Material and methods:

- Survey design:

o The composition of the participating groups and how the consulting experts were recruited should be defined and clarified, as well as specifying how the literature review was conducted.

Thank you for your suggestions. The group was formed by Italian medical residents in Public Health. The participants of the group are the authors of the manuscript and those listed in the acknowledgements. The experts involved were recruited among the universities to which the members of the working group belonged. For what concerns the literature review: the working group brainstormed the mental health dimensions and its determinants to investigate with the questionnaire. Only for those dimensions of mental health for which the willingness to investigate them was unanimously decided in the group, a bibliographic search on Pubmed was carried out. The keywords used were: name of the dimension to be investigated plus the words "questionnaire" and “validation”. The main results of this research were then discussed again in the working group to select the best tool to include in the survey. We have added this information in the manuscript.

o Perhaps it would be appropriate to add to the eligibility criteria of the scales more objective aspects such as the validity and reliability obtained in their validation.

Thank you, of course the selection also comprehended the evaluation of the validity and reliability of the tools. We have now added this information in the text.

o Google Form does not allow you to include terms and conditions, so the information may not be protected, so a more robust data storage system would be preferable.

We thank the Reviewer for having highlighted this point. Since Google Form does not allow automatically including terms and conditions, at the beginning of the questionnaire we set a “question” containing all the information and allowing for only two options: “I give the consent” or “I do not give the consent”. Only those who opt for giving the consent can proceed with the full questionnaire. Participants who disagree are redirected to a thank you message.

Regarding the issue of data protection, we will not use Google Form neither Google Drive as a storing system, but only to spread the questionnaire and to collect data. All the collected data will be then downloaded and stored in a professional computer and protected by a password. This information is reported in the text.

o How long will the survey be available for fulfillment?

The fill out of the survey will be available for about one month. We have added this information, thank you.

o It is not explicit whether adding two ad hoc questions to the International Physical Activity Questionnaire (IPAQ) scale influences its reliability and validity.

The questions added are just exploratory, but don’t have impact on the final score, so that do not affect the questionnaire validity or reliability. Thanks to your suggestion. We specified that in the manuscript.

o At least a preliminary exploratory analysis of the unvalidated Work-Related Stress Questionnaire would be appropriate.

The preliminary exploratory analysis of the Work-related Stress Questionnaire is available at DOI: 10.7417/CT.2020.2235. We added the citation to this article in the paper.

- Sample size:

o Given that the group of resident physicians are mostly young, a young population reference should be used to calculate the sample size, since the PASSI reference used refers to the population between 18 and 69 years of age, age being a factor that may influence the prevalence of Depressive Disorders.

We appreciate your consideration about the age effect on the onset of depression. We decided to calculate the sample size using PASSI surveillance because PASSI gathers data on Italian population aged 18-69, a range that comprehends the age of public health residents. In fact, nowadays, no epidemiologic data on depression among medical residents are available in Italy. After deeper research in the literature, we have found that worldwide prevalence of depression among medical residents is 28.8% according to the results of a recent meta-analysis. Therefore, we recalculated the sample size using this value.

- Ethical considerations, patient information and written informed consent.

o Given the nature of the study and the variables collected, signed consent must be obtained from the participants.

We appreciate this consideration and understand the importance of the ethical aspects of a study which involves human beings. In order to start the completion, as mentioned above, it will be firstly necessary to accept the informed consent on data processing. Without the acceptance of consent, it will not be possible to complete the survey. Furthermore, by clicking on “I give the consent” the system records the acceptance and allows for proceeding with the full questionnaire.

o Clarify whether the data will be encrypted or anonymized.

Thank you for the note. Data will be collected directly in an anonymous form from the survey, with no possibility of identifying participants in any way.

o Clarify, if data collection is to be done online, where the data will be stored and the protection measures in place.

As mentioned above data collection will be carried out with an online survey on Google Form. However, the storage of the survey results will be offline in a password-protected file on a professional computer.

- Discussion:

o The obligation to fill in all the data is mentioned as a strength of the study; however, it is pointed out that gender may not be filled in, which is likely to have an influence as a strength.

The possible answers to the question about gender were: 1) male; 2) female; 3) nonbinary; 4) I’d rather not specify. Therefore, even if there was the obligation to answer the question, the participant could not specify the gender by selecting the fourth option.

- Limitations:

o Among the limitations, it should be noted that although alcohol consumption is explored by means of the AUDIT questionnaire, it does not ask about the consumption of other drugs, which could be highly relevant.

We are grateful to the reviewer for this comment which gave us the opportunity to provide more information on the limitations of the study. A sentence has been added in the Limitations’ section.

Reviewer 2 Report

Thank you for the invitation to review the paper “Public Health Residents’ Anonymous Survey in Italy (PHRASI): study protocol for a cross-sectional study for a multidimensional assessment of mental health and its determinants”.

The study appears interesting and the protocol is promising. I have minor comments that can help to strengthen the quality of this paper.

Introduction

Please make sure to put the references numbers (i.e. [4], [6], [9], etc.) at the end of the sentence rather than in the middle.

Methods

Questionnaire

It appears to me that it needs more than 10 minutes, the authors can perform a pilot study on 5 participants to get the precise needed time.

The Cronbach's alpha of each scale should be calculated among the study sample to verify the reliability of the assessment. 

Sample size

The sample size calculation section should be restructured. The authors cannot use the prevalence of depressive symptoms only to calculate the sample size, as the current study is assessing other mental health outcomes. Moreover, the outcomes are being assessed among public health residents and not the general population. The authors may use the percentage of the medical residents among other residents or students to do the calculation, or find another suitable way for sample size calculation.

Also the formula of Charan and Biswas used for calculation should be stated.

Ethical consideration

The study should be approved by an IRB or institutional ethics committee, even it is observational and data collection won’t trace personal identifiers. The IRB/ ethics committee may waive the need for a written informed consent, yet the study cannot be carried out without ethical approval.

Discussion

I suggest that the authors add a “Practical Implications” section in the discussion before the limitations. It should discuss what will be next, and how the findings of this research will be implicated in practice.

Additional comments

I have a suggestion to the authors for future research, the authors can validate a single tool from their questionnaire that can be used as a short comprehensive assessment for mental health among medical residents.

Author Response

Thank you for the invitation to review the paper “Public Health Residents’ Anonymous Survey in Italy (PHRASI): study protocol for a cross-sectional study for a multidimensional assessment of mental health and its determinants”.

The study appears interesting and the protocol is promising. I have minor comments that can help to strengthen the quality of this paper.

Introduction

Please make sure to put the references numbers (i.e. [4], [6], [9], etc.) at the end of the sentence rather than in the middle.

We are so grateful for your suggestion. The manuscript has been changed accordingly.

Methods

Questionnaire

It appears to me that it needs more than 10 minutes, the authors can perform a pilot study on 5 participants to get the precise needed time.

We performed a pilot study with 6 participants. The mean time for the response was 11,5 minutes (10 minutes, 11 minutes, 15 minutes, 11 minutes, 12 minutes, 10 minutes respectively). We changed the information on the time needed to complete the questionnaire.

The Cronbach's alpha of each scale should be calculated among the study sample to verify the reliability of the assessment.

In the validation studies, when available, Cronbach’s alpha for Patient health Questionnaire-9 is 0.89, for the Italian version of Sick, Control, One, Fat, Food Test is 0.64, for Work-Sense of Coherence Questionnaire is 0.83, for Insomnia severity index is 0.76. We added this information in Materials & Methods. Furthermore, after the completion period, we planned to assess the internal consistency (through Cronbach’s alpha or other instruments) of all the tools used in our questionnaire, using the data collected from participants.

Sample size

The sample size calculation section should be restructured. The authors cannot use the prevalence of depressive symptoms only to calculate the sample size, as the current study is assessing other mental health outcomes. Moreover, the outcomes are being assessed among public health residents and not the general population. The authors may use the percentage of the medical residents among other residents or students to do the calculation, or find another suitable way for sample size calculation.

We appreciate your suggestion very much. Since depression is our main outcome, the sample size was calculated in order to have significant results for this dimension. Nevertheless, since anxiety is our secondary outcome, we added in the manuscript the minimum sample size required to obtain statistically significant results also for this dimension. Furthermore, as suggested, even if there are no studies assessing prevalence of depression and anxiety among medical residents in Italy, we substituted, in the calculation of the minimum sample size required, the prevalence of depression in the Italian general population with its prevalence in medical residents assessed in other studies. In this way we are able to identify a more precise sample size.

Also the formula of Charan and Biswas used for calculation should be stated.

We are grateful for your advice. We have written the formula in the appropriate section of the manuscript.

Ethical consideration

The study should be approved by an IRB or institutional ethics committee, even it is observational and data collection won’t trace personal identifiers. The IRB/ ethics committee may waive the need for a written informed consent, yet the study cannot be carried out without ethical approval.

We appreciate this consideration and understand the importance of having the approval of an ethical committee for any study which involves human beings. On the other hand, we follow the Italian legislation for observational studies without drugs or medical devices, which do not mention the necessity to have an ethical committee approval. Moreover, participation in our study is totally voluntary. The online questionnaire is completely anonymous, and through its answers it is not possible to identify any participant nor harm his/her health or mental status. We added this detailed explanation and the reference to the Italian and European legislation in the manuscript

Discussion

I suggest that the authors add a “Practical Implications” section in the discussion before the limitations. It should discuss what will be next, and how the findings of this research will be implicated in practice.

We thank the Reviewer for having raised this important point that gave us the opportunity to improve the quality of our paper. A “Practical Implications” section has been added in the discussion.

Additional comments

I have a suggestion to the authors for future research, the authors can validate a single tool from their questionnaire that can be used as a short comprehensive assessment for mental health among medical residents.

We thank you for your suggestion. We hope to do this in the next studies.